

# Spatio-Temporal Variations and Uncertainty in Land Surface Modelling for High Latitudes: Univariate Response Analysis

Didier, G Leibovici[1], Shaun Quegan[1], Edward Comyn-Platt[2], Garry Hayman[2], Maria Val Martin[3], Mathieu Guimberteau[4], Arsène Druel[4], Dan Zhu[4], and Philippe Ciais[4]

[1]School of Mathematics and Statistics, University of Sheffield, UK
[2]Centre for Ecology and Hydrology, Wallingford, UK
[3]Leverhulme Centre for Climate Change Mitigation, Animal and Plant Sciences Department, University of Sheffield, UK
[4]Laboratoire des Sciences du Climat et de l'Environnement, Institut Pierre Simon Laplace, France

**Correspondence:** Didier Leibovici (d.leibovici@sheffield.ac.uk)

**Abstract.**

A range of applications analysing the impact of environmental changes due to climate change, *e.g.* geographical spread of climate sensitive infections (CSIs), agriculture crop modelling, *etc.*, make use of Land Surface Modelling (LSM) to predict future land surface conditions. There are multiple LSMs to choose from that account for land processes in different ways and, depending on the application, the choice of LSM and its sensitivity will have different impacts. For useful predictions for a specific application, one must therefore understand the inherent uncertainties in the LSMs and the variations between them, as well as uncertainties arising from variation in the climate data driving the LSMs. This requires methods to analyse multivariate spatio-temporal variations and differences. A methodology is proposed based on multi-way data analysis, which extends Singular Value Decomposition (SVD) to multi-dimensional tables, and provides spatio-temporal descriptions of agreements and disagreements between LSMs for both historical simulations and future predictions. The application underlying this paper is prediction of how climate change will affect the spread of CSIs in the Fenno-Scandinavian and north-west Russian regions, and the approach is explored by comparing Net Primary Production (NPP) estimates over the period 1998-2013 from versions of leading LSMs (JULES, CLM5 and two versions of ORCHIDEE) that are adapted to high latitude processes, as well as variations in JULES up to 2100 when driven by 34 global circulation models (GCMs). A single optimal spatio-temporal pattern, with slightly different weights for the four LSMs (up to 14% maximum difference), provides a good approximation to all their estimates of NPP, capturing between 87% and 93% of the variability in the individual models, as well as around 90% of the variability in the combined LSM dataset. The next best adjustment to this pattern, capturing an extra 4% of the overall variability, is essentially a spatial correction applied to ORCHIDEE-HLveg that significantly improves the fit to this LSM, with only small improvements for the other LSMs. Subsequent correction terms gradually improve the overall and individual LSM fits, but capture at most 1.7% of the overall variability. Analysis of differences between LSMs provides information on the times and places where the LSMs differ and by how much, but in this case no single spatio-temporal pattern strongly dominates the variability. Hence interpretation of the analysis requires the summation of several such patterns. Nonetheless, the three best principal tensors capture around 76% of the variability in the LSM differences, and to a first approximation successively indicate the times and places where ORCHIDEE-HLveg, CLM5 and ORCHIDEE-MICT respectively differ from the other LSMs.





Differences between the climate forcing GCMs had a marginal effect up to 6% on NPP predictions out to 2100 without specific spatio-temporal GCM interaction.

## 1  Introduction

The rise in surface temperatures under global warming is predicted to be most severe in the Arctic, where it is already altering
surface conditions and perturbing ecological systems (Overland et al., 2014). This will have multiple societal impacts, not least on the health of animals and humans (IPCC AR5 WG2 A, 2014). The term climate sensitive infection (CSI) refers to diseases whose epidemiological aspects are driven, at least in part, by climatic factors (McMichael et al., 2006; Ebi et al., 2017; Cayol et al., 2017). In the Arctic, climate change is likely to cause enhanced CSI risk in terms of increased incidence, more frequent outbreaks, geographic spread of existing affected zones, and occurrence of newly affected zones (Pauchard et al.,
2016; Sajanti et al., 2017; Waits et al., 2018) The complex ecology of CSI organisms presents a challenge to modelling and predicting their epidemiology (Ostfeld, 2010; Carvalho et al., 2014; Ruscio et al., 2015; Sormunen et al., 2016; Li et al., 2016; Gilbert, 2016; White et al., 2018). However, such modelling is needed as disease vectors, such as ticks, mosquitoes, badgers and roedeer, which are associated, for example, with Lyme disease (Borreliosis) and Tularemia are expanding their ranges northwards (Jaenson et al., 2012; Jore et al., 2014; Andersen and Davis, 2017; Laaksonen et al., 2017; Blomgren et al., 2018).
Under climate change and human-induced land use changes, fragmentation of the landscape was found to affect Lyme disease (Simon et al., 2014), whilst mosquito abundance was associated with outbreaks of Tularemia in boreal forest (Rydén et al., 2012). CSIs can also be non-vector diseases, as climate change may force increasing proximity and contacts between animals, *e.g.* pestivirus affects mammals (livestock or wild) and so reindeer (Kautto et al., 2012). This could threaten the successful bovine pestivirus eradication programs existing in Scandinavia since the 1990's (Tryland, 2013).

The Nordic Centre of Excellence (NCoE) CLINF, *"Climate change effects on the epidemiology of infectious diseases and the impacts on Northern societies"* (www.clinf.org), is an interdisciplinary project supported by NordForsk (www.nordforsk.org), covering an area extending across Norway, Sweden, Finland and north-west Russia. Its aim is to study how climate change will affect the prevalence of human and animal CSIs and the consequences for Arctic societies. To do so it needs to characterise how a changing climate will change the environmental and societal conditions affecting a range of CSIs in Nordic regions. Besides
predicting environmental changes likely to affect the spread of CSIs, CLINF also aims to gather and generate information on the societal impacts of climate change. Achieving this aim requires tools to model land surface and aquatic changes under climate forcing. This paper focuses on land surface models (LSMs) and the extent to which existing LSMs could provide forecasts useful for the purposes of predicting CSI epidemiology.

An important factor in discussing the predictive value of these models is the variability in their outputs. This variability arises
from two sources: variability in the climate drivers, since there are many Global Circulation Models (GCMs), and differences between LSMs, whose core concepts are similar but with many differences in process representation and parameterisation. This leads to three key questions:

(i)  How does the choice of the GCM affect the CSI-relevant outputs of a given LSM?



(ii) For a given GCM, how different are the CSI-relevant outputs of the different LSMs?

(iii) How do the joint effects of GCM and LSM differences translate into variability in predictions of CSI-relevant quantities?

Addressing these questions requires methods to describe spatio-temporal differences in models, and the first part of this paper describes such methods. The treatment here is relevant to a range of applications and is generic, but the evaluation of the

methods in the latter part of the paper is couched in terms of differences between LSM predictions of Net Primary Productivity (NPP), *i.e.* a single model output variable indicating vegetation activity, hence with relevance to CSI modelling involving changes in habitat for specific vectors, as well as carbon fluxes and ecosystem functioning (Koca et al., 2006; Rafique et al., 2016).

It is important that we quantify the uncertainty in any variable derived from an LSM model as a predictor in CSI modelling,

so that the full uncertainty in the predictions (and associated risk) is available to public health decision-making. Typically, the uncertainty in the predictions from a single LSM is poorly known, and we instead treat the spread in data simulated by a range of leading LSMs as a proxy for this uncertainty. Since Arctic CSIs are the underlying motivation for this work, we only consider LSMs that represent characteristics of Nordic areas, including high latitude processes, vegetation and landscapes. These are CLM5 (the Community Landscape Model version 5) (Lawrence et al., 2019); JULES (the Joint UK Land Environment Simu-

lator) (Clark et al., 2011; Comyn-Platt et al., 2018); and two versions of ORCHIDEE (ORganizing Carbon and Hydrology in Dynamic EcosystEms), ORCHIDEE-MICT (OR_MICT) (Guimberteau et al., 2018) and ORCHIDEE-HLveg (OR_HL) (Druel et al., 2017). The simulated climate data cover the historical period from December 1997 to December 2013, while for JULES we also analysed data from 100-year forecasts to the end of the 21$^{st}$ century under forcing by 34 different GCMs (Comyn-Platt et al., 2018). The specifics of the four models and the driving climate data are briefly described in section 1.2.

Section 2 motivates the use of a multi-way methodology to characterise variations between LSMs, and the essentials of such a methodology are described in Section 3. In Section 4 we use this methodology to analyse the differences between the four selected LSMs, while Section 5 shows how the methodology can be applied directly to differences between the LSMs. The same approach is then used in Section 6 to assess how the choice of a particular GCM affects the NPP predictions from the JULES LSM. Section 7 gives our discussion and conclusions.

**1.1 LSM and ecological modelling aspects relevant to CSI prediction**

Climate change is driving the spread of a range of CSI disease vectors (Zuliani et al., 2015; Andersen and Davis, 2017; Blomgren et al., 2018), so understanding the spatio-temporal distribution and evolution of characteristics, such as habitat suitability of these vectors or reservoirs, is essential. These characteristics can then be used in an ecological model that could be coupled with epidemiological models to estimate future risks of disease incidence, *e.g.* where and when the transmission

risks are likely to be highest. For example, changes in abundance and extent of habitat suitability are important factors to be considered in dynamic landscape epidemiology modelling (Lambin et al., 2010). Changes due to global warming could affect both abundance and geographical extent, and also extend the period of transmission, *e.g.* by affecting the vector life cycle (Rose





et al., 2015). Extreme weather conditions and events may either introduce outbreaks of abundance, thereby increasing the risk of pathogen occurrence in disease vectors, or wipe out a species at a given location.

Under different scenarios of climate drivers, such as the Representative Concentration Pathways (RCPs) developed by the Intergovernmental Panel on Climate Change (IPCC), LSMs can simulate future atmospheric and land conditions that can

be related to vector habitat suitability. Variables simulated by combined climate and land surface models, such as surface temperature, soil moisture, precipitation and land cover, can be used in ecological models or as part of an epidemiological model, *e.g.* for a species distribution model (Booth et al., 2014). However, the predictive uncertainty in these variables may lead to significant uncertainty in the predictions from CSI modelling (Asghar et al., 2016). This paper describes and quantifies the spatio-temporal uncertainty arising from the choice of LSM alone, *i.e.* without assessing its impact on CSI predictions,

but provides an essential component in understanding the uncertainty in any statistical or mathematical predictions of CSI epidemiology and ecology (Beale and Lennon, 2012; Zuliani et al., 2015; Metcalf et al., 2017) that use LSM outputs as predicting variables.

## 1.2 Land Surface Model and Data Description

The four LSMs used in this study, CLM5, two versions of ORCHIDEE (OR_MICT and OR_HL) and JULES, were chosen

because of their high degree of maturity and their ability to model characteristics of Nordic areas, including high latitude processes, vegetation and landscapes. Table 1 summarises these characteristics; details can be found in the references. OR_MICT (Guimberteau et al., 2018) includes major high latitude adaptations, including snow and soil thermal interaction, plant primary productivity constrained by high latitude conditions, and soil carbon stocks with feedback dynamics. OR_HL (Druel et al., 2017, 2019) adapts ORCHIDEE with specific plant functional types (PFTs) such as non-vascular plants (mosses, lichens),

Arctic C3 grass and boreal shrubs. CLM5 (Lawrence et al., 2019) includes permafrost modelling and snow processes, C3 Arctic grass and deciduous boreal shrubs as part of its 15 PFTs (see Appendix B) but no non-vascular plants. The version of JULES (Clark et al., 2011) used here has been extended to be suitable for high latitudes (Comyn-Platt et al., 2018) by including processes such as permafrost-carbon feedbacks (Burke et al., 2017).

For all the LSMs, the initial PFTs were derived from land cover maps. JULES and the two versions of ORCHIDEE use

the land cover product from the European Space Agency Climate Change Initiative (ESA-CCI) (Poulter et al., 2015). The supplementary material to Druel et al. (2017) describes the correspondence between land cover and the added Arctic PFTs. CLM5 uses the Land Use Harmonised data version 2, a product of the Land Use Intercomparison Project (LUMIP) (Lawrence et al., 2016) to define its initial spatial distribution of PFTs. For the historical analyses, the data were re-gridded to the finest grid-spacing, 0.5°E × 0.5°N, by simple disaggregation which introduces a limitation when comparing the LSMs. All analyses

were performed for a sub-area of the CLINF zone between 4.5°E-34.5°E and 58.5°N-70.5°N. Note that the climate forcing data are not the same for the different LSMs (see Table 1) since the LSM data were provided by different modelling groups, each of which uses preferred GCMs. This is unlikely to have any significant impact on the LSM comparisons (see Section 5).





**Table 1.** Summary of the main characteristics of the four LSMs (for details see references) analysed for the historical period 1997-2013 and the forecasts to 2100 with JULES. Acronyms and references for the GCM drivers are given in the associated references to the LSMs.

| Land Surface Model | Initial grid-spacing | Climate driver model (GCM) | High latitude characteristics and processes |
|---|---|---|---|
| OR_MICT | 1°E 1°N | CRUNCEPv8 | Permafrost thaw - snow processes - soil stocks and carbon feedback on soil temperature - impact of severe climatic conditions on plant productivity - 13 PFTs, including Arctic vegetation, but no non-vascular plants, specific Arctic C3 grass, or evergreen shrubs - no vegetation competition |
| OR_HL | 2°E 2°N | CRUNCEPv7 GSWP3v0 | Permafrost thaw - snow processes - 16 PFTs, including non-vascular plants, Arctic C3 grass, evergreen shrubs and deciduous shrubs, vegetation competition |
| CLM5 | 1.25°E 0.94°N | GSWP3v1 | Permafrost thaw - snow processes - 15 PFTs, vegetation competition, Arctic vegetation, but no non-vascular plants |
| JULES | 0.5°E 0.5°N | WFDEI | Permafrost thaw - snow processes - 14 PFTs, vegetation competition, Arctic vegetation, but no non-vascular plants, no Arctic specific C3 grass, no evergreen shrubs |
| JULES (horizon year 2100) | 3.75°E 2.5°N | 34 GCMs with IMOGEN (1.5°C and 2°C targets) | Permafrost thaw - snow processes - 10 PFTs, vegetation competition, Arctic vegetation, but no non-vascular plants, specific Arctic C3 grass or evergreen shrubs |

## 2   Analysing spatio-temporal variations in LSMs

Unpublished analysis within the CLINF project has identified specific variables whose spatio-temporal behaviour is correlated with CSI incidence; these include vegetation activity (here represented by net primary production [NPP]), soil moisture, soil surface temperature, snow cover, precipitation and land cover. This section concerns analysis of how predictions of such
5   variables differ between LSMs.

For a given variable, say NPP, the data simulated by an LSM can be arranged as a 2-way $Spatial \times Temporal$ table, where the $Spatial$ dimension has as many entries as latitude-longitude positions and the $Temporal$ dimension represents monthly values for each year over the period studied. For our dataset, the historical data simulations from December 1997 to December 2013 have 193 monthly entries over the selected zone of 1152 grid-cells. Therefore for the 4 LSMs we get a 3-way
10   $1152 \times 193 \times 4$ data table per variable or a 4-way $1152 \times 193 \times 4 \times 6$ table if we include all the variables given above. Since the LSMs provide NPP for each PFT, the PFT dimension could also be added, but this is not done here.

Analysing such structured datasets to understand spatial, temporal and between-model variations can be challenging when there are long-tail distributions (as is the case in our dataset: see Fig.1, which shows the histogram of NPP values in the

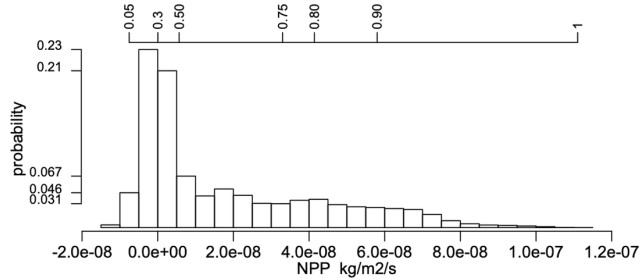

**Figure 1.** Histogram of NPP values (kg m$^{-2}$ s$^{-1}$) in the 3-way table for the 4 LSMs for the period 1998-2013 and the selected CLINF region; the top axis indicates cumulative frequencies.

combined historical datasets simulated by the 4 LSMs) which preclude the use of classical geostatistical methods, and due to their multi-variate nature. For 2-way tables, Singular Value Decomposition (SVD) is a powerful tool to extract associations of variables and patterns within data, *e.g.* clusters and trends. The SVD of a data matrix finds pairs of vectors (components) that successively extract decreasing fractions of the variation in the data and are uncorrelated with previous pairs. Visual description

of these optimal vectors can be obtained by plotting the component weights, *e.g.* a *Spatial* effect as a map and a *Temporal* effect as a time series plot.

With more than two dimensions, combining different dimensions to obtain a 2-way table suitable for SVD would lead to difficulties in interpreting the results. We could instead compare the SVDs of the four spatio-temporal ($1152 \times 193$) tables of NPP for each LSM, which may indicate whether the models behave similarly, but would not readily highlight their differences.

Such considerations have led to the development of methods to extend the SVD to multi-way tables; these are briefly described below, before giving a fuller decription in Section 3 of the PTA$k$ method used in this paper (Leibovici, 2010), which is an optimal nested decomposition of the data variation.

## 3   From Singular Value Decomposition to multi-way data analysis

Let $X$ be a $n \times p$ matrix, which we can regard as a collection of $n$ $p$-dimensional vectors or $p$ $n$-dimensional vectors. The

matrix $X^t X$ is positive semi-definite, so all its eigenvalues are positive, and its eigenvectors, $\varphi_h$, are mutually orthogonal, *i.e.* $\varphi_h^t \varphi_{h'} = 0$ if $h \neq h'$. The matrices $X^t X$ and $X X^t$ have the same eigenvalues, $\sigma_h^2$, and the sum of squares of the elements of $X$ is given by $\boldsymbol{x}^t \boldsymbol{x} = trace(X^t X) = trace(X X^t) = \sum_h \sigma_h^2$, where $\boldsymbol{x}$ is the matrix $X$ vectorised as a $np$-dimensional vector.

The SVD of any matrix $X$ is defined by the series of decreasing $\sigma_h$, the singular values, each associated with a pair of unit vectors $\varphi_h$ ($p$-dimensional) and $\psi_h$ ($n$-dimensional), with $\psi_h^t \psi_h = \varphi_h^t \varphi_h = 1$, which explain a fraction $\sigma_h^2/(\sum_h \sigma_h^2)$ of the

variability of $X$ (defined as the sum of squares of the elements of $X$), *i.e.* $\varphi_h^t X^t X \varphi_h = \sigma_h^2 = \psi_h^t X X^t \psi_h$. Hence SVD can be





used for dimension reduction by defining a $p'$-dimensional subspace ($p' < p$) that captures most of the variability in $X$:

$$
\begin{aligned}
X &= SVD(X) = \sum_{h=1}^{p'} \sigma_h \psi_h \varphi_h^t + \sum_{h>p'} \sigma_h \psi_h \varphi_h^t \\
&= \sum_{h=1}^{p'} \sigma_h \psi_h \otimes \varphi_h + \epsilon \qquad\qquad\qquad (1)
\end{aligned}
$$

For a suitable $p'$, the residual variation $\epsilon = \sum_{h>p'} \sigma_h \psi_h \varphi_h^t$ is small enough to be considered as insignificant. As shown in

equation (1), this decomposition can be written in tensorial form, since $\psi_h \varphi_h^t = \psi_h \otimes \varphi_h$. The rank-1 matrix $\psi_h \varphi_h^t$ is known
as a decomposed rank-1 tensor (Leibovici, 2010). The term $\sigma_1 \psi_1 \otimes \varphi_1$ is the best rank-1 tensor approximation to $X$ in the
sense of capturing the maximum fraction of variability in $X$ among all rank-1 tensors, *i.e.* the maximum value of $\sigma = \psi^t X \varphi$.
Subsequent rank-1 tensors in the decomposition in equation (1), given by the other eigenvectors, are orthogonal to the previous
ones and successively extract decreasing fractions of the variability. Matrices can be seen as order 2 tensors and multi-way

tables as order $k$ tensors, where $k$ is the number of dimensions of the table. For $k = 2$ the SVD can be seen as an optimal basis
vector system in each dimension and in the tensor space, and generalisations of SVD to tensors of order $k \geq 3$ aim to find
equivalent optimal systems.

Several algorithms to extend SVD to tables with more than 2 entries have been proposed (Tucker, 1966; Carroll and Chang,
1970; Harshman, 1970; Kroonenberg, 1983; Leibovici et al., 2007; Leibovici, 2010) and development of methods and their

applications is is still very active (Demšar et al., 2013; Kroonenberg, 2016; Takeuchi et al., 2017; Lock and Li, 2018). Most
extensions aim to find an optimum decomposition of a multi-way table that allows dimension reduction by looking for a
decomposition similar to equation (1) under specific optimisation criteria. For a multi-way table $X$ with $k \geq 3$ entries this
takes the generic form:

$$
\begin{aligned}
X &= SVD\_k\_method(X) + \epsilon \\
&= \sum_{h_1, h_2, \ldots, h_k} \sigma_{h_1 h_2 \ldots h_k} \psi_{h_1} \otimes \varphi_{h_2} \otimes \ldots \otimes \xi_{h_k} + \epsilon \qquad\qquad (2)
\end{aligned}
$$

where the $h_i$ index the basis vectors of the individual vector spaces making up the $k$-dimensional data table and $\epsilon$ expresses the
residual of the approximation given by the summation. This residual depends on the method and the number of components
used in the decomposition, and can be zero (as would be the case if we retained all the terms in a SVD).

The decompositions carried out by the CANDECOMP and PARAFAC methods (Carroll and Chang, 1970; Harshman, 1970)

fix the number of rank-1 tensors in the decomposition but do not impose an orthogonality constraint, while PCA-3modes
(Kroonenberg, 1983) considers both orthogonality and rank within each vector space. However, all three methods need to
choose in advance the number of rank-1 tensors in their optimisation and obtain decompositions that are not nested as with
SVD, in which the rank $p''$ approximation of $X$ contains the approximation obtained for $p'$ (with $p'' > p'$). This property is
often desirable for environmental data analysis (Frelat et al., 2017), as decomposition of the variance or sum of squares has a

practical interpretation.





To address this, Leibovici and Sabatier (1998) developed the PTA$k$ method, which is a hierarchical decomposition giving nested approximation by construction. For $k = 2$, the PTA$k$ algorithm is the same as SVD, while for $k = 3$ it is given by:

$$
\begin{aligned}
X = \mathrm{PTA3}(X) \quad = \quad & \sigma_1(\psi_1 \otimes \varphi_1 \otimes \phi_1) \\
+ \quad & \psi_1 \otimes_1 \mathrm{PTA2}(P_{(\varphi_1 \otimes \phi_1)^\perp} X .. \psi_1) \\
+ \quad & \varphi_1 \otimes_2 \mathrm{PTA2}(P_{(\psi_1 \otimes \phi_1)^\perp} X .. \varphi_1) \\
+ \quad & \phi_1 \otimes_3 \mathrm{PTA2}(P_{(\psi_1 \otimes \varphi_1)^\perp} X .. \phi_1) \\
+ \quad & \mathrm{PTA3}(P_{(\psi_1^\perp \otimes \varphi_1^\perp \otimes \phi_1^\perp)} X) \qquad\qquad\qquad (3)
\end{aligned}
$$

The notation $\otimes_i$ means that the vector on the left takes the $i$th place in the tensor product, *e.g.* $\varphi_1 \otimes_2 (\alpha \otimes \beta) = \alpha \otimes \varphi_1 \otimes \beta$ and ".." indicates the contraction operation (defined in Appendix A along with definitions of the other notation used in equation (3)). Note that the PTA3 algorithm is recursive as the last line of equation (3) calls another PTA3. This process can be continued until it leads to a null table, but normally a stopping rule is imposed by requiring the decomposition to capture a prescribed fraction of the overall variability or specifying the desired number of order $k$ PTs.

Similarly to SVD, the PTA3 algorithm first retrieves the rank-1 tensor approximation to $X$, $\sigma_1 \psi_1 \otimes \varphi_1 \otimes \phi_1$, that captures the maximum possible fraction of the variability in $X$. The second, third and fourth lines in equation (3) correspond to optimisations associated with this first PT in which the decomposed tensors share one of the components in the the first PT. The corresponding PTA2 analyses are complete SVD decompositions into series of tensor products. Given this decomposition, descriptive statistics or plots of the triple of components $(\psi_1, \varphi_1, \phi_1)$ can then be used to visualise the pattern or effect associated with the fraction of the variability captured by each of the tensors.

The generalisation of equation (3) to $k$-way data tables is straightforward. In a PTA$k$ decomposition, the first rank-1 tensor will have associated PTA$(k-1)$'s which will recursively end up at associated PTA2's, *i.e.* SVDs.

## 4 Spatio-temporal variations of NPP across the 4 LSMs

This principal aims of this section are to perform a PTA3 analysis of the 3-way $Spatial \times Temporal \times LSM$ table $X$ of NPP and to interpret the results. However, it is useful to first examine some of the properties of the distributions of NPP for each LSM. The histogram of the NPP values in the full data table $X$, displayed in Fig.1, conceals distinct differences between the LSMs. Some of these differences are indicated by Table 2, which gives the mean NPP and sum of the squares of NPP for each LSM, and Table 3, which shows for each LSM the fraction of NPP values in each decile of the reference distribution in Fig.1. In Table 2, OR_MICT stands out by its low mean NPP (23% less than JULES) and low variability (significantly less than the other LSMs, and 37% less than OR_HL). The LSMs also exhibit different distributions (see Table 3): notably CLM5 has 35% of its NPP values in the first decile of the reference distribution, while OR_HL and JULES have very few values in this decile, and the decile with peak occupancy is different for all four LSMs. However, all the LSMs place around 10% of their NPP estimates in each of the higher deciles (70% to 100%).





**Table 2.** Mean NPP (kg m$^{-2}$ s$^{-1}$) and sum of squares of NPP (SS) for the original aggregated and individual LSMs, together with the SS explained by each PT from the PTA3 analysis, and the cumulative approximations (in brackets) to the overall SS and the SS of each LSM.

|  | OR_MICT | OR_HL | CLM5 | JULES | overall |
|---|---|---|---|---|---|
| mean NPP ($\times 10^{-8}$) | 1.63 | 1.93 | 1.73 | 1.99 | 1.82 |
| SS ($\times 10^{-10}$) | 1.70 | 2.33 | 2.12 | 2.09 | 8.23 |
| mean NPP in PT `-no-1` ($\times 10^{-8}$) | 1.69 | 1.92 | 1.84 | 1.88 | 1.83 |
| PT `-no-1` ssPT % (cumul %) | 92.00 (92.00) | 86.50 (86.50) | 87.30 (87.30) | 93.10 (93.10) | 89.50 (89.50) |
| PT `-no-6` ssPT % (cumul %) | 0.62 (92.62) | 9.36 (95.86) | 2.32 (89.62) | 1.35 (94.45) | 3.72 (93.22) |
| PT `-no-3` ssPT % (cumul %) | 0.68 (93.30) | 0.00 (95.86) | 4.20 (93.82) | 1.95 (96.40) | 1.72 (94.94) |
| PT `-no-9` ssPT % (cumul %) | 1.42 (94.72) | 1.33 (97.19) | 1.35 (95.17) | 1.43 (97.83) | 1.38 (96.32) |
| PT `-no-7` ssPT % (cumul %) | 2.90 (97.62) | 0.04 (97.23) | 0.91 (96.08) | 0.05 (97.88) | 0.86 (97.18) |
| PT `-no-11` ssPT % (cumul %) | 0.00 (97.62) | 0.05 (97.28) | 1.05 (97.13) | 0.50 (98.38) | 0.42 (97.60) |
| PT `-no-4` ssPT % (cumul %) | 1.09 (98.71) | 0.27 (97.55) | 0.01 (97.14) | 0.14 (98.52) | 0.34 (97.94) |
| PT `-no-10` ssPT % (cumul %) | 0.18 (98.89) | 0.17 (97.72) | 0.17 (97.31) | 0.19 (98.71) | 0.18 (98.12) |
| PT `-no-21` ssPT % (cumul %) | 0.01 (98.90) | 0.39 (98.11) | 0.03 (97.34) | 0.21 (98.92) | 0.17 (98.29) |
| PT `-no-16` ssPT % (cumul %) | 0.01 (98.91) | 0.24 (98.35) | 0.01 (97.35) | 0.11 (99.03) | 0.10 (98.39) |

These distributional differences tell us nothing about the spatio-temporal differences between the LSMs, and for that we use the decomposition provided by the R package PTA$k$ (Leibovici, 2010) of which the first ten terms are displayed in Fig.2. This describes the hierarchical and nested decomposition of the sum of squares of $X$ into PTs and associated PTs. Each row corresponds to a PT, identified by a label and number, `-no-`, and its singular value, *e.g.* `vs111` and `-no-1` correspond to the first line of equation (3) giving the best rank-1 approximation of $X$, with singular value $\sigma_1 = 2.7147e - 05$. The row with label `vs222` gives the singular value corresponding to the next order-3 rank-1 approximation, which corresponds to the recursive step in the last line of equation (3).

**Table 3.** Deciles (q) of the reference NPP distribution given by Fig.1 and the percentage of NPP values in each decile observed for each LSM. An LSM whose NPP values had a distribution similar to the reference would have 10% in each decile; entries in bold indicate departures of more than 2% from 10%.

|  | 10% | 20% | 30% | 40% | 50% | 60% | 70% | 80% | 90% | 100% |
|---|---|---|---|---|---|---|---|---|---|---|
| q | -1.25e-09 | -5.15e-11 | 4.58e-11 | 1.30e-09 | 5.67e-09 | 1.53e-08 | 2.65e-08 | 4.14e-08 | 5.80e-08 | 1.11e-07 |
| OR_MICT | **5.2** | **22.0** | 8.0 | **5.6** | 10.0 | **14.0** | 9.7 | 8.9 | **7.2** | 9.4 |
| OR_HL | **0.0** | **4.2** | **21.0** | 12.0 | 12.0 | 11.0 | 11.0 | 9.5 | 8.0 | 11.0 |
| CLM5 | **35.0** | **6.2** | **2.1** | **3.0** | **5.8** | **5.6** | 10.0 | 11.0 | 11.0 | 9.9 |
| JULES | **0.2** | **7.4** | 8.6 | **19.0** | 12.0 | 10.0 | 9.5 | 10.0 | **13.0** | 9.7 |





The rows between `vs111` and `vs222` correspond to PTs associated with `vs111`, which are derived from PTA2s, *i.e.* SVDs. The labels given to these decomposed components start with the dimension of the component used in contracting the tensor $X$ (see Appendix A) and continue with the label of the PT from which they are derived and the dimensions of the contracted tensor, *e.g.* `1152 vs111 193 4` identifies the results from the PTA2 of the $193 \times 4$ matrix $X..\psi_1$ (*i.e.* an SVD), where $\psi_1$

5  is the 1152-dimensional vector forming the *Spatial* component of PT `-no-1`. Therefore the associated PTs `-no-3` and `4` have the same *Spatial* component as tensor `-no-1`. Similarly, the rank-1 tensors `-no-6` and `7` are associated PTs along the *Temporal* component of `vs111`. Note that Fig.2 displays only PTs with a contribution exceeding $0.1\%$ of the total sum of squares, as indicated in the bottom line in the figure. This means that we show only the first two terms from each of the PTA2s associated with `vs111`, one of the associated PTs associated with `vs222`, and no associated PTs for `vs333`. The other terms

```
++++ PTA-  3 modes ++++    Spatial x Temporal x LSM
                  data =    1152        193         4
                   ssX = 8.2338e-10
      ---Percent Rebuilt from Selected --- 98.39 %
                   -no-   -SingVal-      -ssPT %
vs111                 1   2.7147e-05      89.50
1152 vs111 193 4      3   3.7596e-06       1.72
1152 vs111 193 4      4   1.6700e-06       0.34
193 vs111 1152 4      6   5.5311e-06       3.72
193 vs111 1152 4      7   2.6545e-06       0.86
4 vs111 1152 193      9   3.3694e-06       1.38
4 vs111 1152 193     10   1.2108e-06       0.18
vs222                11   1.8484e-06       0.42
193 vs222 1152 4     16   9.1985e-07       0.10
vs333                21   1.1894e-06       0.17
++++                 ++++
   Selected over sum of squares (ssPT)> 0.1 % total
```

**Figure 2.** Summary of the PTA3 decomposition for the data table of NPP simulations for the 4 LSMs for the studied region and period. Each line of the table corresponds to a rank-1 tensor part of the decomposition; the variability (sum of squares) in $X$ it explains is given by the square of its singular value, `SingVal`, and this is expressed as a percentage of the variability in the `ssPT %` column.

10  in the rows of Fig.2 are the singular values associated with each PT (`SingVal`) and the percentage of the variability in $X$ explained by each of the PTs (`ssPT %`). The variability explained is given by the square of the singular value. Tensors `-no-` 2, 5 and 8 are missing as they are repeats of already listed rank-1 tensors. This arises from the way the code implements equation (3); see Leibovici (2010) for further details.

The contribution by the main PTs decreases from `vs111`, `vs222`, `vs333`, *etc*. Each of the associated tensors makes a

15  smaller contribution than its main PT but this may be larger than the next main PT, *e.g.* tensor `-no-3` captures more variability than tensor `-no-11`. There is no particular ordering in the tensors associated with different components, between `-no-3` which is associated with the *Spatial* component and `-no-6` which is associated with the *Temporal* component, but the PTs associated with a given component are ordered since they derive from the same PTA2 (*i.e.* SVD), *e.g.* `-no-3` precedes `-no-4`. Fig.2 then allows one to select the PTs or associated PTs that successively capture the variability in $X$.

20  It is helpful to visualise the first PT, whose components are displayed in Fig.3, as an optimal approximation to the initial $1152 \times 193 \times 4$ data table in which each of the 4 layers is the same 2-D spatio-temporal "map", but scaled by the weight for





a particular LSM, given by $\phi_1$. The spatial pattern at each time is the same ($\psi_1$, as in Fig.3(a)), but with a weight appropriate to that particular time. Similarly, the time series at each spatial location is the same ($\varphi_1$, shown as Fig.3(b)), but with a weight appropriate to that location. To recover the NPP from this approximation at a particular position, time and for a given LSM, the corresponding values in $\psi_1$, $\varphi_1$ and $\phi_1$ are multiplied together and then multiplied by its singulat value, $\sigma_1$. Exactly the same
construction applies to each of the rank-1 tensors in the decomposition.

The *Spatial* effect (Fig.3(a)), which is always positive, places higher weights in Sweden, the Baltic states and north-west Russia, and lower values in Norway and northern Finland, with values varying between $22\%$ and $138\%$ of a uniform *Spatial* weighting (*i.e.* equal weights of $1/\sqrt{1152}$). For display, the *Temporal* component, a vector of length 193 (December 1997 to December 2013), has been split into annual segments which express the monthly weights over the 16 year period (Fig.3(b)).
As expected, there is a strong seasonal effect, with the summer months (June to August) having large positive weights, while values are very small from November to March and include negative values from December to February in nearly all years. Two other groups of months can be distinguished: October paired with April as just before or after winter, and May with September as just before and after the seasonal peak of production. The months from May to September all display significant upward trends in NPP over the 16 years, with average increases of $1.48\%$, $0.80\%$, $0.63\%$, $0.67\%$ and $0.51\%$ per annum respectively.
The other months show no significant trends. April, May, June and August have more inter-annual variability than the other months, and April, May and June all show peaks in 2002. Over the 16 years, the maximum is in July 2013 and is $217\%$ greater than for uniform temporal weighting ($1/\sqrt{193}$), while the minimum in winter (December 2006) represents $-8\%$ of uniform weighting.

Since these spatial and temporal patterns are the same for all the LSMs, the difference between them is expressed by the
LSM weights (Fig.3(c)). For identical LSM simulations, the weights would be $1/2$, since each vector in the decomposition is normalised to unity (*i.e.* $\sqrt{\phi_{11}^2 + \phi_{12}^2 + \phi_{13}^2 + \phi_{14}^2} = \sqrt{4\phi_{11}^2} = 1$), but the weights lie between $0.460$ and $0.523$, with JULES and OR_HL respectively giving values $3\%$ and $5\%$ higher than for equal weights, and OR_MICT giving a value $8\%$ lower. Hence there is only a weak dependence on the LSM in this first PT. The proportion of the variability in the first PT due to each LSM is given by the squares of the LSM weights, *i.e.* $21.2\%$, $27.4\%$, $25.1\%$ and $26.3\%$ for OR_MICT, OR_HL, CLM5 and
JULES, respectively. Multiplying these values by $\sigma_1^2$ gives the sum of squares of NPP in the spatio-temporal maps for `vs111` for each LSM (see Table.2). Several points should be noted about the approximation to $X$ given by `vs111`:

1. The squares of the LSM weights are in the ratio $1 : 1.29 : 1.19 : 1.24$, while the values of the original sum of squares of NPP (see Table 2) are in the ratio $1 : 1.37 : 1.25 : 1.29$. Hence the first PT correctly picks up the ordering of the variability amongst the LSMs, but not its full value, since it is effectively a smoothing of the dataset.

2. The spatio-temporal maps for the individual LSMs capture $92.0\%$, $86.5\%$, $87.3\%$ and $93.1\%$ of the original variability of OR_MICT, OR_HL, CLM5 and JULES, respectively. Hence each one is a reasonable approximation to the original LSM simulation, particularly for OR_MICT and JULES.

3. The mean NPP represented by $vs111$ is $1.834 \times 10^{-8}$ kg m$^{-2}$ s$^{-1}$, which is very close to that of the mean of $X$ ($1.824 \times 10^{-8}$ kg m$^{-2}$ s$^{-1}$), though the individual NPP spatio-temporal maps for each LSM track the original mean

**Figure 3.** Plots of the components of PT -no-1 of the PTA3 decomposition in Fig.2 representing 89.5% of the variability.

NPP less closely (+3.6%, -0.7%, +6.3% and -5.6% for OR_MICT, OR_HL, CLM5 and JULES respectively; see Table 2).

As noted above, recovering the NPP at a particular position, time and for a given LSM in $vs111$ requires multiplying together the corresponding weights in the $Spatial$, $Temporal$ and $LSM$ dimensions and then multiplying by the singular value. So, for example, the maximum value of NPP in the first PT over the whole time-period is in July 2013, in the darkest red cell of





Fig.3(a) and for OR_HL, the LSM with maximum weight. Since $\sigma_1 = 2.7147 \times 10^{-5}$ and in this cell the $Spatial$, $Temporal$ and $LSM$ weights are 0.040, 0.156 and 0.523, respectively, this yields a maximum NPP of $8.9 \times 10^{-8}$ kg m$^{-2}$ s$^{-1}$. There are small negative $Temporal$ weights from December to February, leading to negative values of NPP and an overall minimum NPP of -1.44 $\times 10^{-8}$ kg m$^{-2}$ s$^{-1}$ in December 2006, which will again occur for OR_HL and at the same position as the overall

maximum NPP.

The second best PT in the decomposition, $-no-6$, is a $Temporal$-associated PT, so has the same $Temporal$ component as $vs111$, and expresses $3.72\%$ of the variability. Its $Spatial$ component (Fig.4(a)) has positive (red) weights in the north and west and negative (green) weights to the south and east. The most striking feature of this tensor is in its $LSM$ component (Fig.4(c)) which shows a marked contrast between OR_HL, with a large negative weight, and the other LSMs, for which the

weights are significantly smaller and positive. Hence, after multiplying the weights in the different components, all the LSMs except OR_HL will see an increase in NPP in the red areas in the summer months and reduce it in the green areas, while the opposite effect occurs for OR_HL. When the $Temporal$ weights are negative, as occurs for most of the winter, these sign changes in NPP are reversed. As can be seen from Table 2, including the contribution from this PT increases the captured fraction of variability in OR_HL from 86.5% to 95.9%, with much smaller gains for the other LSMs.

Fig.5 shows the components of the third best PT, $-no-3$ which captures $1.72\%$ of the variability and is associated with the same positive $Spatial$ pattern as PT $-no-1$. Here the $Temporal$ effect is positive for the months from August to October, close to zero for November and July, and negative for the other months, especially April to June. CLM5 and JULES have large positive and negative weights respectively while OR_MICT has a smaller negative weight and OR_HL has a weight which very close to zero. Hence for CLM5 this tensor acts to increase NPP from August to October and reduce it for all other months

except November and July, while for JULES and OR_MICT it does the opposite. As expected from the weights, including this tensor mainly acts to improve the fit of CLM5 and JULES to their original values (Table 2).

Principal tensor $-no-9$ is the fourth best in the decomposition and captures $1.38\%$ of total variability. Since it is associated with the LSM component of PT $-no-1$ it is the same for all LSMs. Its $Spatial$ component Fig.6(a) exhibits a strong latitudinal gradient with positive values in the north and negative values in the south. The $Temporal$ component has positive weights in

July and August and negative values in April, May and June, while for other months the weights are near zero. Hence, since the LSM weights are all positive, in July and August this tensor acts to increase NPP in the north and reduce it in the south, while in April to June it does the opposite. These effects will be slightly greater for OR_HL because of its greater weight. Though its contribution to the overall sum of squares is only $1.38\%$, it provides improvements for all LSMs (see Table 2).

None of the other PTs contributes more than 1% to the overall variability and their components are not displayed, although

the contributions for all terms in Fig.2 are given in Table 2. For example, the next best PT ($-no-11$), which derives from the last line in equation (3), captures $0.42\%$ of the variability and principally improves the fit to the variability captured by CLM5 and, to a lesser extent, JULES. The summation of all 10 PTs that each represent at least $0.1\%$ of the variability captures 98.4% of the variability in $X$ and between 97.4% and 99.0% of the variablity in the individual LSMs (last line of Table 2).

Overall, this analysis shows that a single optimal spatio-temporal pattern, with slightly different weights for the four LSMs

(up to 14% maximum difference), provides a reasonably good approximation to all their estimates of NPP, capturing between





(a) Spatial dimension

(b) Temporal dimension                    (c) LSM dimension

**Figure 4.** Plots of the components of PT `-no-6` associated with PT `-no-1` along its $Temporal$ dimension, which is therefore identical to Fig.3(b); it represents 3.72% of the variability.

87% and 93% of the variability in the individual models, as well as around 90% of the variability in the combined LSM dataset. The next best adjustment to this pattern is a spatial correction that principally applies to OR_HL and significantly improves the fit of the approximation to this LSM, with only small improvements for the other LSMs. The second best adjustment adds a temporal pattern that mainly affects CLM5 and JULES and improves the fit to these LSMs, with less effect on OR_MICT and none on OR_HL. The third best adjustment adds a new spatio-temporal pattern whose spatial component is roughly the

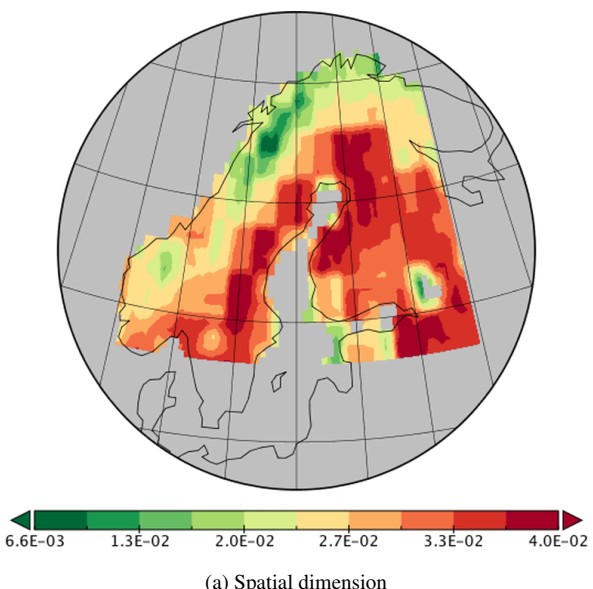

(a) Spatial dimension

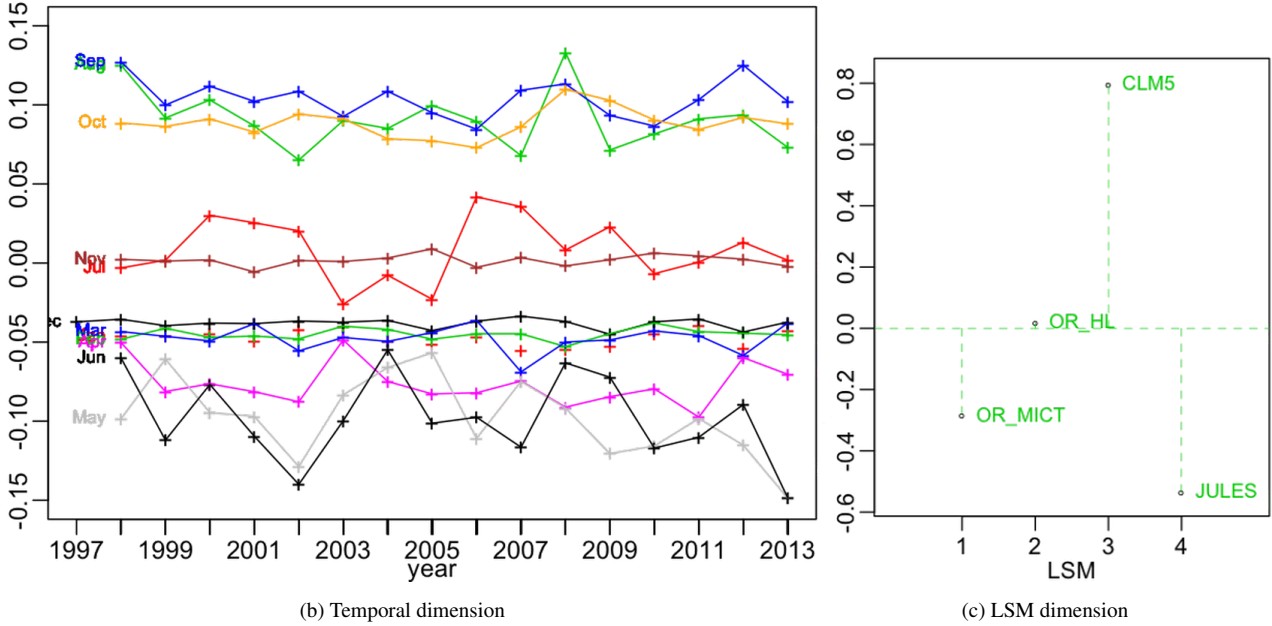

(b) Temporal dimension                    (c) LSM dimension

**Figure 5.** Plots for PT -no-3 associated with PT -no-1 along the *Spatial* dimension, which is therefore identical to Fig.3(a)); it represents 1.72% of the variability.

opposite of that in the first PT (*i.e.* it is spatially similar but with opposite signs) but a quite different temporal component that is positive in the later summer months, negative in the late spring and early summer months, and roughly zero at other times. . The improvement in the overall fit from the next best PT and all succeeding ones is less than 0.9%, and, although in two instances the fits to individual models improve by over 1%, in most cases the improvements are much smaller (see Table 2).



(a) Spatial dimension

(b) Temporal dimension

(c) LSM dimension

**Figure 6.** Plots for PT `-no-9` associated with PT `-no-1` along the $LSM$ dimension, which is therefore identical to Fig.3(c)); it represents 1.38% of the variability.

Summing the 10 PTs whose individual contribution to the overall variability exceeds 0.1% (Fig.2) provides an approximation to the overall data table that captures 98.4% of the overall variability and between 97.4% and 99.0% of the variability in the individual LSMs (Table 2). However, also of interest is the point-wise goodness of fit of the approximation, not just the variability it captures. This is represented by the table of residuals, *i.e.* the $\epsilon$ term in eq. (2). Around 75% of the absolute values of these residuals are less than 8.4% of the overall mean NPP, so in most cases there is a good point-wise fit to the original





data, but the maximum absolute value of the residuals ($4.83 \times 10^{-8}$) is around the third quartile of NPP ($3.44 \times 10^{-8}$). Hence, in some cases the approximation may be significantly different from the correct value despite the residuals contributing less than $1.62\%$ to the overall variability.

## 5 Analysing differences between the LSMs

Section 4 identified differences between the LSMs captured by an optimal decomposition of the associated 3-way table. In this section we instead directly analyse the variability in the differences between the LSMs, in order to localise where and when the LSMs disagree and thus to quantify spatio-temporally the uncertainty in NPP associated with the choice of a particular LSM. We in fact analyse LSM differences normalised by the maximum value of NPP, *i.e.* $(\mathrm{NNP}_1 - \mathrm{NPP}_2)/\mathrm{NPP}_{max}$, where $\mathrm{NNP}_1$ and $\mathrm{NNP}_2$ refer to NPP values in two different LSMs and $\mathrm{NPP}_{max}$ is the maximum NPP over all 4 LSMs. Note that for each

pair of LSMs we have chosen arbitrarily whether to use ($\mathrm{NPP}_1$ - $\mathrm{NPP}_2$) or ($\mathrm{NPP}_2$ - $\mathrm{NPP}_1$). This choice of sign does not affect the PTA$k$ optimisation since this is based on the sum of squares, but the sign does matter when identifying which of a pair of LSMs gives higher NPP values. The sign convention used is indicated on the relevant figures (Figs.9-11).

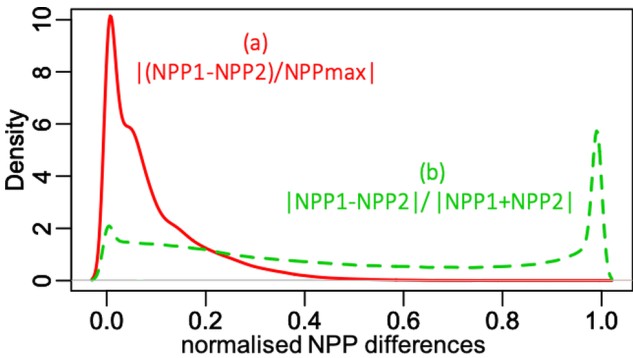

**Figure 7.** Histograms of (a) the absolute values of the 6 NPP differences normalised by $\mathrm{NPP}_{max}$ and (b) the normalised relative differences.

Fig.7 displays the histograms of (a) the absolute values of normalised differences, which has a peak near zero but also a fairly long right hand tail, and (b) the absolute values of ($\mathrm{NPP}_1$ - $\mathrm{NPP}_2$)/($\mathrm{NPP}_1$ + $\mathrm{NPP}_2$) which is fairly flat across most of the

range, with a small peak near zero, but with a large peak peak near 1. The latter indicates that for many times and places the NPP values in one LSM are very small relative to one of the other LSMs. This occurs much more frequently in winter when CLM5 gives NPP values that are very small compared to those from the other LSMs. However, since NPP is small in winter, these large relative differences have little impact on overall annual production. Indeed Table 2 shows that the mean annual NPP from CLM5 exceeds that from OR_MICT.

The results of the PTA3 for the $1152 \times 193 \times 6$ table of normalised NPP differences are shown in Fig.8. The first and second PTs respectively extract $43.4\%$ and $21.7\%$ of the variation, both with well-structured patterns in their components. The first, shown in Fig.9, has a spatial pattern with negative (green) values areas to the south and east and positive (red) values in the





north and west, as well as south-east Finland. The $Temporal$ component is always positive and displays a seasonal effect (Fig.9(b)) with the same ordering of the months as Fig.3. All the differences involving OR_HL have significant weights but for the other differences they are close to zero. Hence the effects of this principal vector essentially translate into differences between OR_HL and the other LSMs. Taking into account the signs of the $Spatial$, $Temporal$ and $LSM$ weights (the last to

5   be interpreted as an LSM difference), this means that for this PT over the whole time period CLM5 > JULES > OR_MICT > OR_HL in the red areas in Fig.9(a) while these orderings are reversed in the green areas. However, the small weights on the differences not involving OR_HL indicate that the other LSMs all give similar NPP values.

```
++++ PTA-  3 modes ++++ Spatial x Temporal x diff_LSM
                data =   1152        193          6
                ssX =  2.9111e-10
    ---Percent Rebuilt from Selected --- 93.29 %
                     -no-  -SingVal-      -ssPT %
vs111                  1    1.1237e-05     43.37
1152 vs111 193 6       3    1.4257e-06      0.70
1152 vs111 193 6       4    6.0759e-07      0.13
193 vs111 1152 6       6    5.6423e-06     10.94
193 vs111 1152 6       7    3.7595e-06      4.85
6 vs111 1152 193       9    2.7450e-06      2.60
6 vs111 1152 193      10    2.3891e-06      1.96
vs222                 11    7.9543e-06     21.73
1152 vs222 193 6      13    2.8396e-06      2.77
193 vs222 1152 6      16    8.5840e-07      0.25
6 vs222 1152 193      19    2.2970e-06      1.81
6 vs222 1152 193      20    1.3750e-06      0.65
vs333                 21    1.4961e-06      0.77
6 vs333 1152 193      29    1.1445e-06      0.45
vs333 1152 193      30    9.5587e-07      0.31
++++                  ++++
Selected over sum of squares (ssPT)> 0.1 % total
```

**Figure 8.** Summary of the PTA3 decomposition for the data table of the 6 normalised LSM differences.

The second best PT, Fig.10, expressing 21.7% of the variability, has a quite complex positive spatial pattern, with the strongest effects in northern Finland and the weakest near Lake Ladoga, Russia, in the south-east of the region. The temporal

weights are positive in June, May and to a lesser extent April, weakly positive for the winter months from December to March, nearly zero in July, variable but mainly negative in November, and negative from August to October. The weights for all differences involving JULES are negative but are positive for the other differences. This means that for this PT in all locations and for all years JULES > OR_MICT > OR_HL > CLM5 from April to June, but this ordering is reversed from August to October.

The third and fourth most important PTs, -no-6 and 7, are associated with the $Temporal$ component of vs111 and capture 10.94% and 4.85% of the variability, respectively. Their $Spatial$ and $LSM$ components are depicted in Fig.11. The first displays little spatial structure apart from significant negative values along the east coast of Sweden. This may be due to differences in data resolution before grid transformation but also occurs where C3 grass is the dominant PFT (all LSMs). All LSM differences have positive weights except CLM5 – JULES, which is negative but small, and all differences involving

OR_MICT have significantly larger weights than other combinations. Since the temporal component is everywhere positive

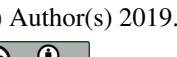



(a) Spatial dimension

(b) Temporal dimension

(c) LSM dimension

**Figure 9.** Best PT (vs111) of the PTA3 decomposition of the 6 normalised differences, representing 43.37% of the variability. In (c), the labelling CLM5_JUL indicates the difference CLM5-JULES, and similarly for other LSM pairs.

(Fig.9(b)), the net effect is that OR_MICT > OR_HL > JULES > CLM5 in the red areas (with a high value north of Lake Ladoga), and this order is reversed in the green areas. However, the differences between LSMs other than those involving OR_MICT are small.





(a) Spatial dimension

(b) Temporal dimension

(c) LSM dimension

**Figure 10.** Second best PT (`vs222`) of the PTA3 decomposition of the 6 normalised differences, representing 21.7% of the variability.

The *Spatial* component of PT `-no-7` (Fig.11(c)) is weakly positive except for a very small area near Tromsö in northern Norway. All the LSM differences involving JULES have negative weights and have greater magnitude than the other differences, which are all positive, meaning that JULES > OR_HL > OR_MICT > CLM5 everywhere except near Tromsö, where this ordering is reversed.







(a) Spatial dimension

(b) LSM dimension

(c) Spatial dimension

(d) LSM dimension

**Figure 11.** *Spatial* and *LSM* components of PTs $-$no$-6$ and 7 associated with the *Temporal* component of vs111 in the PTA3 decomposition of the 6 normalised differences, representing $10.94\%$ and $4.85\%$ of the variability, respectively.

As indicated by Fig.8, the next best PTs (not displayed) are $-$no$-13$, associated with the *Spatial* component of $vs222$, $-$no$-9$ and 10, associated with the LSM component of $vs111$, and $-$no$-19$, associated with the LSM component of $vs222$. Hence PT $-$no$-13$ modulates the temporal pattern of differences depicted in Fig.10 with a distinct temporal pattern that has different positive weights for each of the LSM differences (the contribution from OR_HL - CLM5 is almost zero and OR_MI – JULES gets the larger positive weight. A contrast between July (positive weights) and May (negative weights) stands out clearly





from the other months by the size of its contribution to the variability, for reasons which are not clear. In Fig.10, July and and OR_MI – JULES weights were close to zero. Because PTs `-no-9` and `10` are associated with the LSM component of $vs111$, the spatio-temporal table given by summing the $Spatial$ x $Temporal$ terms in all three PTs can be analysed together; this would mainly reveal spatio-temporal differences between OR_HL and the other LSMs (see Fig.9(c)). However, this combined

analysis cannot be displayed as separate $Spatial$ and $Temporal$ plots. With the same LSM weights as in Fig.10, PT$-no-19$ exhibits a clear north-south gradient and a temporal pattern in which June clearly contributes more to the variability than the other months. This is similar to what is seen for July in PT $-no-13$, again for unknown reasons. All the rest of the PTs cumulatively contribute only 10% to the overall variability and individually less than 0.8%.

Also analysed was the variability in the quantity $|NPP_1 - NPP_2| \, / \, |\, NPP_1 + NPP_2 \,|$ but this is not displayed, since its main

contribution is to show that the large peak near 1 seen in Fig.7 (plot (b)) can mainly be attributed to small values of CLM5 relative to the other LSMs in winter in the north of the region.

The analysis in this Section adds significantly to that in Section 4 by providing specific information on the times and places where the LSMs differ and by how much. However, in this case no single spatio-temporal pattern strongly dominates the variability so interpretation of the analysis requires consideration of several such patterns. Nonetheless, the three best PTs

capture around 76% of the variability in the LSM differences. The first essentially tells us that over a well-defined spatial pattern and a clearly ordered temporal pattern that with a maximum in summer and a minimum in winter, OR_HL gives different values from the other LSMs, which are all similar. The second PT principally identifies times and places where CLM5 differs from the other LSMs, while the third does the same for OR_MICT.

## 6   Climate forcing uncertainty

This section analyses the effects of different GCM drivers on the NPP estimated by JULES, so is a partial answer to question (i) in Section 1. Two global warming scenarios that stabilise at 1.5°C and 2.0°C above pre-industrial levels by year 2100 were used, with 34 GCMs as climate forcing in JULES (Comyn-Platt et al., 2018). The ensemble of the GCMs is taken to represent the uncertainty in climate prediction, from which one can get an idea of the associated uncertainty in the JULES estimates of NPP. Note however, that this commonly-used approach to quantifying climate uncertainty is not entirely satisfactory, since

it identifies inter-GCM model variability with the internal uncertainty in climate prediction (Hawkins and Sutton, 2009; Kay et al., 2015).

For each scenario a PTA3 analysis was performed on a $Spatial \times Temporal \times GCM$ table. The decompositions for both the 1.5°C and 2.0°C targets capture almost all the variation in their first PT (99.15% and 99.16% respectively), hence very similar spatio-temporal patterns of NPP are produced whichever GCM is used. The spatial patterns are shown in Figs.12(a)

and 13(a). The temporal and GCM weights are given as a percentage relative deviation from uniform weighting, *i.e.* $100\times$ (cp – unif)/unif, where cp indicates the weight while unif $= 1/\sqrt{1200}$ for the $Temporal$ dimension and unif $= 1/\sqrt{34}$ for the GCMs.





Over the 100 years, all months exhibit an initial increase, which is sharper for the 2°C scenario, followed by a flattening out and minor decrease; this decrease sets in around 2070 for the 1.5°C scenario and slightly later for the 2.0°C scenario. The maximum increase from 2000 (indicated on each monthly curve in Figs. 12(b) and 13(b)) is higher in every month for the 2.0°C scenario, *e.g.* 20% and 32% in July for the 1.5°C and 2.0°C case, respectively. The differences between the GCMs are indicated

by histograms of the relative deviation of the GCM weights from uniform weighting (Figs. 12(c) and 13(c)). These differences are up to 7% for the 2.0°C scenario and 4.5% for the 1.5°C scenario. For both scenarios, the groups of GCMs giving lowest or highest difference from equal weighting was the same, though the precise ordering was different (see Appendix C). If the singular value associated with this first PT are expressing the same amount of variability, the latter is 10% higher for the 2.0°C case than for 1.5°C, which simply expresses the sharper increase of NPP values produced under a more intense warming.

## 7  Conclusions

This paper investigates the uncertainty associated with choosing a given LSM and GCM to predict the effects of climate change on Net Primary Production in northern Europe. More precisely, it provides a spatio-temporal analysis that captures the principal similarities and differences between LSM estimates of NPP, which need to be taken into account if these LSMs are to be used to provide scenarios for applications. Its primary motivation is to provide information relevant to studying Climate

Sensitive Infections (CSIs), but here the CSI context is used only to reduce the number of LSMs to those that contain adequate descriptions of key high latitude processes. It is based on a methodology that extends the SVD of a matrix to a multi-table in order to analyse spatio-temporal variations between LSMs. This allows quantification of the differences between the LSMs and the variability arising from using different climate forcing models (GCMs) when estimating NPP.

Global statistical differences were found between the LSMs, with OR_MICT exhibiting significantly lower mean NPP

and variability than the other LSMs, and CLM5 producing a very high proportion of low values associated with the winter season, particularly in the north of the CLINF region. However, all the LSMs tend to agree for higher NPP values (above the 70% decile), which mainly indicates that they give similar values in summer. Despite these global differences, to a first approximation the spatio-temporal behaviour of all the LSMs could be well-fitted by the tensor product of a single spatial and temporal pattern, in which the west and north of the region exhibited lower NPP values than the east and south, and there

was a strong seasonal pattern. Differences between LSMs for this single pattern were fairly small, with weights lying between 92% and 105% of an uniform weighting of 0.5 or 14% maximum difference between them. This combined pattern captured around 90% of the overall variability in simulations covering 16 years for the whole Fenno-Scandinavian region. Across this time-period, this first approximation displayed statistically significant increases in NPP from May to September in , with the largest increase in the earlier months. This is likely to be caused by the growing season starting earlier and lasting longer.

The LSM requiring most adjustment to this first approximation for an improved fit was OR_HL; this adjustment is in the spatial pattern, decreasing the spatial weights in Norway and northern Finland and increasing them in Sweden and southern Finland. The next adjustment, which has no effect on OR_HL, is to modify the temporal pattern; this particularly improves the fit to CLM5. The approximation achieved with just these adjustments captures 95% of the overall variation and between







(a) Spatial dimension 1.5°C

(b) Temporal dimension 1.5°C

(c) GCM dimension 1.5°C

**Figure 12.** Components of the best principal tensor from the PTA3 analysis of NPP for JULES driven by 34 different GCMs (using IMOGEN) under the +1.5°C target scenario. For the $Temporal$ and $GCM$ dimensions the percentage relative difference from uniform weighting, $100 \times (cp – unif)/unif$, of the component weights is plotted, where 'cp' and 'unif' refer to component weights and uniform weighting respectively. On the $Temporal$ plot the increase for each month between 2000 and the maximum value is indicated as an absolute increase above the 2000 value. The GCM weights are shown as a histogram but individual weights are given in Appendix C.





(a) Spatial dimension 2°C

(b) Temporal dimension 2°C

(c) GCM dimension 2°C

**Figure 13.** Components of the best principal tensor from the PTA3 analysis of NPP for JULES driven by 34 different GCMs (using IMOGEN) under the +2°C target scenario. For the $Temporal$ and $GCM$ dimensions the percentage relative difference from uniform weighting, 100 x (cp – unif)/unif, of the component weights is plotted, where 'cp' and 'unif' refer to component weights and uniform weighting respectively. On the $Temporal$ plot the increase for each month between 2000 and the maximum value is indicated as an absolute increase above the 2000 value. The GCM weights are shown as a histogram but individual weights are given in Appendix C.





93% and 96% of the variation in the individual models. It also has the advantage of being fairly simple to interpret because OR_HL dominates the first adjustment while CLM5 (and to a lesser extent JULES) dominates the second. Further terms in the approximation yield smaller gains that tend to be spread more evenly across the LSMs.

While the first analysis provides information on temporal and spatial patterns characterising the LSMs, more specific in-

formation on how they differ is gained by analysing their differences. Here no single pattern dominates the overall variability between the LSMs, but the three best PTs capture around 76% of the variability in the LSM differences, and can be fairly well interpreted in terms of how individual LSMs differ in space and time from the others. Successively they show where and when individually OR_HL, CLM5 and OR_MICT differ from the other LSMs, and also where different LSMs agree.

Our analysis of the impact of the choice of GCM on the simulations of NPP was restricted to runs with JULES out to 2100

driven by 34 different GCMs. This showed that a single spatio-temporal pattern captured over 99% of the variability of NPP in the combined dataset for climate change scenarios leading to either 1.5°C or 2.0°C atmospheric warming, and that none of the GCM weightings differed by more than 3% from uniform weighting (maximum difference of 6%). The temporal pattern showed increases of NPP up to the 2070s, with small decreases thereafter. Although this analysis was only carried out for JULES, there is no reason to expect different findings for the other LSMs.

Returning to the three key questions posed in Section 1:

(i) How does the choice of the GCM affect the CSI-relevant outputs of a given LSM?

(ii) For a given GCM, how different are the CSI-relevant outputs of the different LSMs?

(iii) How do the joint effects of GCM and LSM differences translate into variability in predictions of CSI-relevant quantities?

the analysis in this paper suggests that, at least for NPP, we can neglect the effect of different GCMs and need only deal

with question (ii). Quantitative answers are provided to this question both in terms of spatio-temporal patterns and differences and similarities of LSMs. However, we have only considered one of the six variables listed at the start of Section 2 that are considered to be of major importance for Climate Sensitive Infections (CSIs), and may find different behaviour for the others. In particular, initial investigation indicates very different representations of land cover between the four LSMs and how land cover will evolve under climate change in the 21st century. This variable is likely to be the one showing most differences

between the LSMs because it is very much controlled by the PFTs used, how they are parametrised, and the rules by which PFTs compete over time.

Of significant interest would be analysis of multiple variables and their co-variation. We intended to address this issue in a future paper using the PTAk method used here, since this can be readily extended to multiple variables. While this does not present any methodological difficulties, it will only become clear how useful this is when we find how easy it is to interpret the

outputs of the analysis.

The next major step is to couple the findings from this paper (and its extension to other variables) to ecological models for CSI vectors and statistical epidemiological models in order to establish the sensitivity of predicted CSI behaviour under climate change to the choice of GCM and LSM. Currently only a small number of CSIs have well-developed predictive models





(notably tularemia (Rydén et al., 2012; Andersen and Davis, 2017; Desvars-Larrive et al., 2017); Lyme disease (Simon et al., 2014; Li et al., 2016)) and these will provide the basis for such a study. However, CLINF is in process of developing more comprehensive statistical CSI models at high latitudes, which will lend themselves readily to combination with the approach adopted in this paper.

*Code and data availability.* The analyses were performed using the R package PTA$k$ (https://CRAN.R-project.org/package=PTAk). The multi-way data tables used in the paper can be requested from the first author. CLM5.0 is publicly available through the Community Terrestrial System Model (CTSM) git repository (https://github.com/ESCOMP/ctsm); all model data are archived and publicly available at the UCAR/NCAR Climate Data Gateway, https://doi.org/10.5065/d6154fwh.

## Appendix A: Contraction operator and orthogonal projector

### A1  Contraction

For $X$ and $Y$ two multi-way data tables $n \times p \times q$, their inner product is defined as $<X, Y> = \sum_{ijk} X_{ijk} Y_{ijk}$. The contraction operation .. is the extension to tensors of the linear combination of the columns or rows of a matrix to give a vector. If $X$ is a tensor of order 3, equivalent to a table $n \times p \times q$, then with the variables $(u, v, w)$, vectors of length $n$, $p$ and $q$, respectively, the contraction $X..u$ is a $p \times q$ matrix with $(X..u)_{jk} = \sum_i X_{ijk} u_i$, the contraction $X..v$ is a $n \times q$ matrix with
$(X..v)_{ik} = \sum_j X_{ijk} v_j$, and $X..w$ is a $n \times p$ matrix with $(X..w)_{ij} = \sum_k X_{ijk} w_k$. Contacting $X$ successively by two vectors gives for example $(X..u)..v = \sum_{ij} X_{ijk} u_i v_j = \sum_{ij} X_{ijk} (u \otimes v)_{ij} = X..(u \otimes v)$ and $X..(u \otimes v \otimes w)$ is equivalent to the inner product for the multi-way data tables.

### A2  Orthogonal projector

Without loss of generality let $u$, $v$ and $w$ be unit vectors of dimensions $n$, $p$ and $q$ respectively. If $X$ is a tensor represented
by an $n \times p \times q$ array, one can write $X = (a \otimes b \otimes c)\beta + \epsilon = P_{(a \otimes b \otimes c)}X + P_{(a \otimes b \otimes c)\perp}X$, where $P_{a \otimes b \otimes c} = (a \otimes b \otimes c)\beta$ is the linear orthogonal projection of $X$ onto $a \otimes b \otimes c$ and $P_{(a \otimes b \otimes c)\perp}X = X - (a \otimes b \otimes c)\beta$. From the orthogonality constrains, $\beta = X..(a \otimes b \otimes c)$, so $P_{a \otimes b \otimes c}X = (a \otimes b \otimes c)X..(a \otimes b \otimes c)$.

Moreover, if $X = (x \otimes y \otimes z)$ then $P_{a \otimes b \otimes c}X = P_a x \otimes P_b y \otimes P_c z$. This property extends easily to any subspace of $E$, $F$, and $G$, i.e $P_{E_1} \otimes P_{F_1} \otimes P_{G_1}$ is equivalent to $P_{E_1 \otimes F_1 \otimes G_1}$.

## Appendix B: List of plant functional types (PFTs) used in the LSMs

This appendix lists the PFTs for the versions of the LSMs used in this paper (see section 1.2). JULES, ORCHIDEE_MICT (OR_MICT), ORCHIDEE-HL-Veg (OR_HL) and CLM5 have 14, 13, 16 and 15 PFT PFTs, respectively. The version of JULES used for the 34 simulations over 100 years used 10 PFTs (where C3 or C4 crops or pastures are set as C3 or C4 grass).



## Appendix C: GCM's Weightings from the analysis in section 6

The acronyms of the 34 GCM are derived from the information given in "Table SI.1 CMIP5 Models considered for inclusion in the IMOGEN ensemble" in the supplementary information of the paper Comyn-Platt et al. (2018).

*Author contributions.* EC-P, GH, MVM, MG, AD, DZ and PC provided data and comments on the draft manuscript. DGL designed the
5 methodologies, performed the analyses and drafted the manuscript. SQ and DGL finalised the article.

*Competing interests.* The authors declare that they have no conflict of interest.

*Acknowledgements.* This work was carried out under NordForsk funding to CLINF, a Nordic Centre of Excellence (NCoE) led by Pro-
fessor Birgitta Evengård (*https://www.nordforsk.org/en/programmes-and-projects/projects/climate-change-effects-on-the-epidemiology-of-
infectious-diseases-and-the-impacts-on-northern-societies-clinf* ) under Grant Agreement no. 76413. Maria Val Martin was supported by
10 the Leverhulme Trust through a Leverhulme Research Centre Award (RC-2015-029).



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





**Table B1.** Table of plant functional types relevant in high latitude calculations for each LSM

| PFTs & other tiles bor=boreal tem=temperate | JULES | OR_MICT | OR_HL | CLM5 |
|---|---|---|---|---|
| Bare ground | 1 | 1 | 1 | 1 |
| Broadleaf Deciduous | 1 | 1 tem, 1 bor | 1 tem, 1 bor | 1 |
| Temperate Broadleaf Evergreen | 1 | 1 | 1 | 1 |
| Needleleaf Deciduous | 1 | 1 | 1 | 1 |
| Needleleaf Evergreen | 1 | 1 | 1 | 1 |
| Deciduous Shrubs | 1 | 1 | 1 bor, broadleaf) | 1 bor, 1 tem |
| Evergreen Shrubs | 1 | 0 | 1 bor broadleaf | 1 tem |
| C3 grass | 1 | 1 | 1 | 1 |
| C4 grass | 1 | 1 | 1 | 1 |
| C4 pasture | 1 | 1 | 1 | 1 |
| Urban (tile) | 1 | 1 | 1 | 1 |
| Inland water (tile) | 1 | 1 | 1 | 1 |
| Land ice (tile) | 1 | 1 | 1 | 1 |
| C3 Arctic grass | 0 | 0 | 1 | 1 |
| Non-vascular plants | 0 | 0 | 1 | 0 |





**Table C1.** Rounded GCM component weights relative to uniform weighting from Fig.12 and Fig.13

| GCM acronym | 1.5° | 2° |
|---|---|---|
| CEN_CMCC_MOD_CMCC-CMS | -2 | -3 |
| CEN_CSIRO-QCCCE_MOD_CSIRO-Mk3-6-0 | -2 | -2 |
| CEN_IPSL_MOD_IPSL-CM5A-MR | -2 | -3 |
| CEN_MPI-M_MOD_MPI-ESM-LR | -2 | -2 |
| CEN_MPI-M_MOD_MPI-ESM-MR | -2 | -3 |
| CEN_BCC_MOD_bcc-csm1-1 | -1 | -1 |
| CEN_CNRM-CERFACS_MOD_CNRM-CM5 | -1 | -2 |
| CEN_INM_MOD_inmcm4 | -1 | 0 |
| CEN_MIROC_MOD_MIROC-ESM-CHEM | -1 | -1 |
| CEN_MIROC_MOD_MIROC-ESM | -1 | -1 |
| CEN_NASA-GISS_MOD_GISS-E2-R-CC | -1 | 0 |
| CEN_NASA-GISS_MOD_GISS-E2-R | -1 | 0 |
| CEN_NCAR_MOD_CCSM4 | -1 | -2 |
| CEN_NSF-DOE-NCAR_MOD_CESM1-BGC | -1 | -1 |
| CEN_BCC_MOD_bcc-csm1-1-m | 0 | -1 |
| CEN_BNU_MOD_BNU-ESM | 0 | 0 |
| CEN_CCCma_MOD_CanESM2 | 0 | -1 |
| CEN_IPSL_MOD_IPSL-CM5A-LR | 0 | -1 |
| CEN_MRI_MOD_MRI-CGCM3 | 0 | 1 |
| CEN_NASA-GISS_MOD_GISS-E2-H | 0 | 0 |
| CEN_CSIRO-BOM_MOD_ACCESS1-0 | 1 | 1 |
| CEN_CSIRO-BOM_MOD_ACCESS1-3 | 1 | 1 |
| CEN_MOHC_MOD_HadGEM2-CC | 1 | 1 |
| CEN_MOHC_MOD_HadGEM2-ES | 1 | 1 |
| CEN_NASA-GISS_MOD_GISS-E2-H-CC | 1 | 1 |
| CEN_NOAA-GFDL_MOD_GFDL-CM3 | 1 | 1 |
| CEN_NOAA-GFDL_MOD_GFDL-ESM2M | 1 | 1 |
| CEN_NSF-DOE-NCAR_MOD_CESM1-CAM5 | 1 | 1 |
| CEN_NSF-DOE-NCAR_MOD_CESM1-WACCM | 1 | 1 |
| CEN_IPSL_MOD_IPSL-CM5B-LR | 2 | 2 |
| CEN_MIROC_MOD_MIROC5 | 2 | 3 |
| CEN_NCC_MOD_NorESM1-M | 2 | 3 |
| CEN_NCC_MOD_NorESM1-ME | 2 | 3 |
| CEN_NOAA-GFDL_MOD_GFDL-ESM2G | 2 | 3 |