# Peer review of "Spatio-Temporal Variations and Uncertainty in Land Surface Modelling for High Latitudes: Univariate Response Analysis"

_Biogeosciences, 2019_

## Referee Comment (RC1) · Anonymous Referee #1 · 13 Aug 2019

General comments

The authors present an analysis of spatio-temporal agreement and differences between predicted Net Primary Production (NPP) from four land surface models (LSMs). The main method is an extension of singular value decomposition to multi-dimensional matrices/tensors. The authors consider how the method provides a dimensionality reduction of the data to provide Principal Tensors (PT) which represents the dominant patterns in the data set. The manuscript is generally well written with a good description of the application and about the right level of information on the four LSM models.

I think the balance between the description of the method, presentation of the results,

and the interpretation of the results. First, I'm not sure whether such a lengthy description of SVD and then the PTA k method is required for the intended audience. Inconsistent naming also complicates the methods section, for example, referring to the data set as a "data table" and then to as "tensors". The term Principal tensor (PT) is also not introduced in the methods (e.g Pg 8 L12) but is referred to throughout the text. Introducing PT as the extension of the singular vectors to higher dimensions around Pg 8 L12 would help clarify this.

More importantly, the manuscript would benefit from a greater focus on interpreting the PTs and how these explain "uncertainty" in LSMs. There is not much reference to uncertainty in the manuscript. In the presentation of the results there is a good description of the spatial patterns in each PT but not the interpretation of what each PT and it's singular value means for the uncertainty arising from the different LSMs. For example, that one of the PTs is essentially a "spatial correction applied to ORCHIDEE-HLveg" provides information on the biases between LSMs but this point doesn't come through very strongly. The discussion about this (Pg13 L10) makes the point that OR_HL has a temporal evolution which contrasts to the other three LSMs and this discussed somewhat in the conclusions again. But so the question is this a bias in OR_HL or a representation of the uncertainty (in terms of the spread) in the temporal evolution of NPP?

Similarly, if the first principal component explains around 90% of the variance in the data and it has "only a weak dependence on the LSM", I assume that there is quite good agreement between the LSMs and therefore the uncertainty (i.e. spread) arising from different LSMs isn't that large? If that's true, I think considering the LSM-weights for each PT provides more information on the spatio-temporal patterns of uncertainties. Would this reveal whether the main patterns are driven by the difference in LSMs (weights) or other variances (annual variability in the forcing data for example)?

Section 5 focuses on more on the differences between the LSMs by repeating the method on normalised NPP differences which provides the link to the relative spread

of the NPP estimates. But I still think the manuscript is missing that final link between what the PTs explain about different types of uncertainties (i.e bias and precision) and their relative importance (as indicated by the explained variance?). Some of these inferences are found throughout the manuscript but a dedicated section for relating the PTs to uncertainties (in the most common sense as the precision and any biases of an estimate) would help this.

Relatedly, it would be interesting to know how much more informative the method is for inferring this information on uncertainty than something comparatively simpler such as the spatial and temporal patterns in the coefficient of variation (or some other dispersion metric). This could potentially be just a qualitative comparison to show how the method provides more information about the LSM differences than a quantity which is more commonly considered an "uncertainty".

Finally, a greater discussion of how the PTs and information could be used for quantifying uncertainty for CSI models would be a good addition. For example, does the method essentially provide a representation of the data (in terms of uncertainty ie mean and covariance matrix) which could be used with these models? Or would the original NPPl data have to be propagated through the CSI models to provide uncertainties (i.e. ensemble-based or some moment propagation method)?

Specific comments and minor corrections

Pg.1 L.5 " ...will have different impacts". Impacts on what?

Pg 6 L 2 multi-variate → multivariate

Pg.11 L.4 singulat → singular

Pg.26 L13 Although this analysis was only carried out for JULES, there is no reason to expect different findings for the other LSMs". Why not?

---

## Author Comment (AC1) · 14 Aug 2019

Thank you very much for these comments and questions. ( see the document related to Referee #1)

**1: ". . .lengthy description of SVD and then PTA-k . . ."**

One of the goals of the paper is to describe a methodology that readers could reuse on their data. Therefore, it was necessary to explain enough the generalisation / extension from 2-way data table using the SVD to 3-way or k-way (k>3) data tables with the PTAk. This part of the paper covers in fact 2 pages (bottom of page 6 to bottom of page 8)

which is not that long (2 out of 27 without the Appendix and References). It includes the very minimum and necessary.

**1:" . . .inconsistent naming . . ."**

The naming is not inconsistent but expressing how observed data are arranged into tables that mathematically can be interpreted as tensors. It is the same distinction between a data table with 2 dimensions and a matrix. We will make this more explicit in explaining that tensors are algebraic extension of matrices to more than 2 entries in the corresponding data table.

**1:"The term Principal tensor (PT) is also not introduced . . ."**

Yes Principal Tensor (PT) should be defined at least at Line12 page 8. Thank you for pointing that out. Line 13 is in fact defining it without naming it; only rank-1 tensors with maximum possible fraction of variability in X is used. These rank-1 tensors are the Principal Tensors, in equation (3) The end of Line12 will be changed to ". . .or specifying the desired number of order k rank-1 tensors, or Principal Tensors (PT)."

**1:" . . .the manuscript would benefit from a greater focus on interpreting the PTs and how these explain "uncertainty" in LSMs. "**

Here uncertainty is taken as variation from one LSM to another. Each PT describes at the same time the expected LSM variation spatio-temporally and also differences between the LSMs so the uncertainty in LSM simulations (as expressed by these 4 LSM). (see also your comment about LSM-weights)

**1: " The discussion about this (Pg13 L10) makes the point that . . ."**

The discussion, page13 Line10, is representing a uncertainty from choosing one LSM or another and here choosing ORCHIDEE-HL instead of one of the other LSMs. Is it due to a bias in OR_HL or a lack of accurate modelling from the others? It is difficult to know or to identify exactly?

[Figure]

**1: "Similarly, if the first principal component explains around 90% of the. . ."**

The first PT expressing a large amount of variability, 90%, allows a good approximation of each LSM and you are right the LSM weights are expressing the uncertainty from one LSM to another. For this PT this is up to 14% difference (lowest weight to highest weight) and this can be however relatively important.

We will add a sentence in section 2 on how to relate to spatio-temporal effect in interaction with the uncertainty in LSMs; when LSM-weights are not too different this shows a common feature otherwise it shows uncertainty due to LSM differences (potentially in interaction with an underlying feature existing in LSM)

**1: "Section 5 focuses on more on the differences between the LSMs by repeating the method on normalised NPP differences . . .."**

To be able to distinguish between bias and precision more data would be needed to analyse on one particular LSM.

**1: "Relatedly, it would be interesting to know how much more informative. . ."**

The purpose of the method is to identify spatio-temporally where LSMs are different, i.e. a spatio-temporal representation. Computing coefficients of variations for the 4LSMs would help to understand if they are different but not where and when or for which specific spatio-temporal pattern. Table 2 and 3 provide some example of simpler comparisons.

**1: "Finally, a greater discussion of how the PTs and information could be used for quantifying uncertainty for CSI models . . ."**

Uncertainty propagated to a CSI model is of course in the background as the initial motivation but the paper focuses more on how to describe independently the uncertainty in LSMs. Depending on the disease looked at, the spatial or the temporal may be more important to look at; scale will be also a major factor. This is another challenging issue. Knowing where and when models (and how much) LSMs disagree will be nonetheless

useful when running a CSI model.

**1: Pg.1 L.5 " ...will have different impacts". Impacts on what?**

It is implicitly referring to the outcome of the applicative domain and impact in terms of decision-making error du to uncertainty in the LSM data. We will clarify by adding . . ." impacts in terms of uncertainty in decision-making."

**1: Pg 6 L 2 multi-variate → multivariate**

Thanks. This will be corrected.

**1: Pg.11 L.4 singulat → singular**

Thanks. This will be corrected.

**1: Pg.26 L13 Although this analysis was only carried out for JULES, there is no reason to expect different findings for the other LSMs". Why not?**

Yes, this seems a bit of a leap of faith but the fact that the findings are related by a 1 single PT expressing 99%, so a strong effect which is very coherent with the 1st PT of the analysis with the 4LSM is inclining to make the statement. We will add these arguments in supporting the statement with a sentence like: ". . .here is no reason to expect different findings for the other LSMs (as the findings from a strong effect, 99% of variability are coherent with the first PT in the analysis with the 4 LSMs in Figure 3)"

---

## Short Comment (SC1) · 18 Sep 2019

This is a complementary comment to our previous reply on the point below raised by referee #1.

**1: " The discussion about this (Pg13 L10) makes the point that . . ." "The discussion about this (Pg13 L10) makes the point that OR_HL has a temporal evolution which contrasts to the other three LSMs and this discussed somewhat in the conclusions again. But so the question is this a bias in OR_HL or a representation of the uncertainty (in terms of the spread) in the temporal evolution of NPP? "**

[Figure]

First reply of the 14th of Aug 2019:

The discussion, page13 Line10, is representing a uncertainty from choosing one LSM or another and here choosing ORCHIDEE-HL instead of one of the other LSMs. Is it due to a bias in OR_HL or a lack of accurate modelling from the others? It is difficult to know or to identify exactly?

Added comment 18th of Sept 2019:

Like for GCM modelling, their spread, for any output variable, is usually taken as the uncertainty. It is similar here. The source of this uncertainty is multiple and modelling uncertainty is one of them which contains model adequacy and error propagation (now related to input uncertainty).

---

## Short Comment (SC2) · 27 Nov 2019

I meant partial review, i.e. reviewer #1

———————————————————

---

## Author Comment (AC2) · 27 Nov 2019

[revised manuscript text omitted]

---

## Referee Comment (RC2) · Anonymous Referee #2 · 31 Dec 2019

The main goal of the paper is to estimate possible uncertainties caused by choosing LS and GC models to predict the climate effects on Net Primary Production (NPP) of vegetation cover in northern Europe. Some aspects of the LS model application to describe climate sensitive infections (CSI) are also discussed. The land surface - atmosphere interaction under climate changes is a very important direction of modern studies in ecology, biogeochemistry and meteorology. To solve the key scientific problems in the study authors used four various LS models as well as projections of future climate changes provided by consortium of global circulation models (GCMs). For data processing the multi-way data analysis methods are used. The paper consists of several parts including introduction with study motivation, detailed method part,

the result chapters with sufficient description of the research achievements, conclusion with a summary of the main results. The discussion of obtained results in the paper is relatively short and unfortunately poorly presented. In particular the explanations of found differences in NPP provided by various LS models and their interpretation are not sufficient. Authors consider mainly geographical aspects of the model differences but avoid considering their temporal variations (from year to year). All these questions, I guess, should be considered in the discussion chapter. Another point is related to key objectives and research tasks of the paper. In fact, I cannot clearly see the general study idea and an "internal linkage" of the chapters describing modeling aspects relevant to CSI and NPP predictions. I see some mosaic of separate individual tasks that have to be more closely linked to one another in the paper.

---

## Author Comment (AC3) · 29 Jan 2020

Please see below all the replies to 1 and 2. A pdf file showing the changes from versions R0 to R1 of the manuscript is uploaded as supplementary file.

**................ Replies to Reviewer 1**

Thank you very much for these comments.

**1: "...lengthy description of SVD and then PTA-k ..."**

One of the goals of the paper is to motivate and describe a methodology that readers

could reuse on their own data. Therefore, it was necessary to explain enough the generalisation / extension from 2-way data table using the SVD to 3-way or k>3 way data tables with the PTAk. This part of the paper covers in fact 2 pages (bottom of page 6 to bottom of page 8) which is not that long (2 out of 27 without the Appendix and References). We had reduced it to the very minimum, much of which is devoted to explaining the key equations (1) and (2).

**1:" . . .inconsistent naming . . ." The naming is not inconsistent but expressing how observed data are arranged into tables that mathematically can be interpreted as tensors. It is the same distinction between a data table with 2 dimensions and a matrix. 1:"The term Principal tensor (PT) is also not introduced . . ."** Agreed, Principal Tensor (PT) should be defined at least at Line12 page 8. Thank you for pointing that out. Line 13 was defining it without naming it; only rank-1 tensors with maximum possible fraction of variability in X or is used. These rank-1 tensors are the Principal Tensors. The end of Line12 will be changed to ". . .or specifying the desired number of order k rank-1 tensors, or Principal Tensors (PT)."

**1:" . . .the manuscript would benefit from a greater focus on interpreting the PTs and how these explain "uncertainty" in LSMs. "**

Here uncertainty is taken as variation from one LSM to another. Each PT describes at the same time the expected common LSM variation spatio-temporally and also differences between the LSMs so the uncertainty in LSM simulations (as expressed by these 4 LSM). Quantifying real uncertainty is a much harder problem as this would mean identifying uncertainty of the input (climate data) and sensitivity of each LSM, then each LSM would express its own uncertainty that could be compared, perhaps also compared to observed/measured values of NPP (calibrated etc.). The differences could report on sensitivity, biases and complexity attributed to each LSM. In first instance, under the hypothesis of each chosen LSM being representative of the best models available, the variations or variability from one LSM to another expresses the uncertainty arising from multiple factors.
**1: " For example, that one of the PTs is essentially a "spatial correction applied to ORCHIDEE-HLveg" provides information on the biases between LSMs but this point doesn't come through very strongly. The discussion about this (Pg13 L10) makes the point that OR_HL has a temporal evolution which contrasts to the other three LSMs and this discussed somewhat in the conclusions again. But so the question is this a bias in OR_HL or a representation of the uncertainty (in terms of the spread) in the temporal evolution of NPP? "**

The discussion, page13 Line10, is representing uncertainty from choosing one LSM or another and here the discussion is particularly focussing on ORCHIDEE-HL which to which PT n°6 brings 9.36% variability whilst bringing 0.62%-3.72% to the others. Is it due to a bias in OR_HL or a lack of accurate modelling from the others? It is difficult to know or to identify exactly? It is for sure a specificity of OR_HL highlighted by the decomposition but as we do not have a ground truth, it can only be classified as part of the uncertainty between the 4 LSMs.

After the text in page 13: " As can be seen from Table **??**, including the contribution from this PT increases the captured fraction of variability in OR_HL from 86.5% to 95.9%, with much smaller gains for the other LSMs.", we've added the sentences "*Therefore, PT n°6, mostly contributing to fitting OR_HL (9.36% of variability), is highlighting a specificity relatively to the others. Without ground truth, one cannot tell if this is a bias or a better modelling than the other LSMs and just express a well defined uncertainty.*"

**1: "Similarly, if the first principal component explains around 90% of the variance in the data and it has "only a weak dependence on the LSM", I assume that there is quite good agreement between the LSMs and therefore the uncertainty (i.e. spread) arising from different LSMs isn't that large? If that's true, I think considering the LSM-weights for each PT provides more information on the spatio-temporal patterns of uncertainties. Would this reveal whether the main patterns are driven by the difference in LSMs (weights) or other variances (annual variability in the forcing data for example)? "**

The first PT expressing a large amount of variability, 90%, allows a good approximation of each LSM and you are right the LSM weights are expressing the uncertainty /differences from one LSM to another. For this PT this is up to 14% difference (lowest weight to highest weight) and this can be however relatively important.

We've added the following: *"Results from table 2 and this first PT makes the important point that a single spatio-temporal pattern does well at capturing the NPP from the four LSMs. Whilst this expresses a common trend between the LSM, their weights similarity is up to 14% differences, showing a variation in intensity from one to another. Despite similar photosynthesis modules in most LSMs, parameter settings, such as the choice of PFTs together with different climate datasets (GCM, see Table) and settings in other modules induce these variations. The subsequent PTs provide a series of corrections to this common pattern, expressing LSM specificities, such as how the PFTs are parameterised "*

**1: "Section 5 focuses on more on the differences between the LSMs by repeating the method on normalised NPP differences which provides the link to the relative spread of the NPP estimates. But I still think the manuscript is missing that final link between what the PTs explain about different types of uncertainties (i.e bias and precision) and their relative importance (as indicated by the explained variance?). Some of these inferences are found throughout the manuscript but a dedicated section for relating the PTs to uncertainties (in the most common sense as the precision and any biases of an estimate) would help this."** To be able to distinguish between bias and precision more data would be needed to analyse on one particular LSM. As expressed in a previous reply, bias analysis would mean knowing or defining from another source (measures) the "real world" and precision would require sensitivity experiments for each LSM (at least an uncertainty analysis based on knowing the uncertainty of the input and multiple sampling, e.g. Monte Carlo experiment). Therefore we do not think it is helpful to couch the discussion this way. Hence, as said above and in the introduction to the paper, uncertainty is here used to

indicate variability between models, which does represent a form of uncertainty from the point of view of a user of the data, but does not match the usual statistical metrics of uncertainty. In fact, this would be a very different paper, perhaps looking only at one LSM versus some observed/measured NPP.

**1: "Relatedly, it would be interesting to know how much more informative the method is for inferring this information on uncertainty than something comparatively simpler such as the spatial and temporal patterns in the coefficient of variation (or some other dispersion metric). This could potentially be just a qualitative comparison to show how the method provides more information about the LSM differences than a quantity which is more commonly considered an "uncertainty".**

The purpose of the method is to identify spatio-temporally where LSMs are different. Computing coefficient of variations for the 4LSMs would help to understand if they are different but not where and when or for which specific spatio-temporal pattern. Table 2 and 3 provide some example of simpler comparisons but we point out there that these global statistical measures tell us nothing about spatio-temporal patterns of LSM difference

**1: "Finally, a greater discussion of how the PTs and information could be used for quantifying uncertainty for CSI models would be a good addition. For example, does the method essentially provide a representation of the data (in terms of uncertainty ie mean and covariance matrix) which could be used with these models? Or would the original NPP data have to be propagated through the CSI models to provide uncertainties (i.e. ensemble-based or some moment propagation method)?"**

Uncertainty propagated to a CSI model is of course in the background as the initial motivation but the paper focuses more on how to describe independently the uncertainty in LSMs. Depending on the disease looked at the spatial or the temporal may be more

important to look at; scale will be also a major factor.

Knowing where and when models (and how much) LSMs disagree will be useful when running a CSI model. However, knowing that a single spatio-temporal pattern with slightly different weightings for the four models captures 90% of the overall variability in NPP, with similarly high values for each individual LSM, gives confidence that the choice of the LSM is not critical for CSI modelling (at least when NPP is a relevant).

We have added text to the discussion to make this point, see *changes*.

**1: Pg.1 L.5 " ...will have different impacts". Impacts on what?** It is implicitly referring to the outcome of the applicative domain and impact in terms of decision-making error du to uncertainty in the LSM data. We clarified by rewording the sentence to" There are multiple LSMs to choose from that account for land processes in different ways and this may introduce predictive uncertainty when LSM outputs are used as inputs to inform a given application. "

**1: Pg 6 L 2 multi-variate → multivariate** Thanks, corrected.

**1: Pg.11 L.4 singulat → singular** Thanks, corrected.

**1: Pg.26 L13 Although this analysis was only carried out for JULES, there is no reason to expect different findings for the other LSMs". Why not?** This claims follows from the findings: a 1 single PT expressing 99

We have modified the text to: " *Although this analysis was only carried out for JULES, one could expect similar findings for any LSM since they all use a form of the Farquhar photosynthesis model to derive Gross Primary Production, of which some fraction is allocated to NPP. Moreover, this single PT expressing 99% of variability, highlights a strong effect correlated temporally to the findings of the first analysis with the 4 LSMs (Fig. 3). Hence the insensitivity of the simulated NPP to the choice of GCM is likely to be repeated in the other LSMs. "*

**................. Replies to Reviewer 2**
Thank you very much 2 for all these comments. The revised final version for R1 takes into account both 1 and 2 referees comments and replies. The added file shows the differences from R0, the changes made to the manuscript (in blue or red).

Here are comments and replies to reviewer 2 comments.

**2: "The main goal of the paper is to estimate possible uncertainties** . . . . . . **with sufficient description of the research achievements, conclusion with a summary of the main results."** Thank you very much for this accurate summary of our paper and its findings. The estimation of possible uncertainties is however more about quantification and spatio-temporal description (e.g. the patterns of differences between the LSMs) than estimation of the uncertainty as a global magnitude in the predictions of NPP. To do so the multiway analysis method proposed is also very much one of the goal of the paper.

**2: "The discussion of obtained results in the paper is relatively short and unfortunately poorly presented."** The discussion part is relatively short because it seems more natural to discuss the detailed results within each section. This perhaps under-liesÂăthe comment about the paper feeling like a "mosaic of separate individual tasks" (see reply below). Hence the main discussion tries to give a global picture without going into the details of the results. However, in the revised version of the main discussion we have expanded a little on the overall implications of the earlier parts of the paper in terms of what the methodology is capable of doing and how it would benefit CSI (and other) applications.Âă

(see changes in the file highlighting the differences R1 R0)

**2: "In particular the explanations of found differences in NPP provided by various LS models and their interpretation are not sufficient. Authors consider mainly geographical aspects of the model differences but avoid considering their tem-**

**poral variations (from year to year). All these questions, I guess, should be considered in the discussion chapter."**

Explanations and interpretations of differences from one LSM to another in NPP is a very challenging task. The paper proposes a methodology to be able to highlight where and when they are similar and where and when they are not. These could perhaps be related to specific behaviours of each model, e.g. specific differences in PFT parameters but a great knowledge of each LSM, their modelling choices and impact of fine tuning parameters would be required to find proper explanations and interpretations. However, understanding which sets of parameters specifically are related to a specific pattern in the results might also be a challenging task. This is out of scope of this paper and could be also very difficult to achieve due to the complexity involved. The focus on the data in disclosing where and when the LSM are relatively similar or not, is of practical interest to describe the uncertainties in using one LSM or another and in fine to warn CSI modellers when using LSM output variables (for example).

The methodology analysis in the paper decomposes overall model variability into a series of coupled spatial and temporal patterns, each captured by a principal tensor, and the various figures all show both the spatial and temporal components of these patterns. In addition, the text does in fact discuss temporal aspects of the data. For example, Fig. 3 illustrates the expected strong seasonal effect and its evolution and trends over the 16 years period and aspects of the temporal pattern are described (e.g. different trends per month over the 16 years, p. 11).Âă For the projection to 2100, the year to year trends,Âăsimilarities and differences for the 1.5°C and 2°C scenarios are also described in the second paragraph of p. 23. We suspect that this comment may relate to the reviewer expecting all discussion and interpretation to appear in the discussion section, whereas in our view it is more natural to deal with these temporal effects in the sections describing the results. Nonetheless, on pages 13, 14 and 18 we have added several remarks about the temporal behaviour of the patterns embodied in the PTAk decompositions.

However, we added elements of discussion (in analyses sections and in the discussion, conclusion section) concerning year to year variations in the temporal patterns which were mostly seasonal with relative stability over the 16 years. So, trends and variability from year to year are seen more as the natural variability as most of the time they are not modifying too much the seasonal/monthly patterns. Nevertheless, the year to year variability modulate the levels of uncertainties linked to the effects described.

We have now raised and detailed these aspects in the main discussion including any year to year effect to be noticed or not. This includes some apparent singularities (when describing the principal tensors) within the year to year variability (in association with their spatial pattern and LSM difference pattern) e.g.: -in Fig. 5 more between year variations from May to July than other months (post peak productions months Sep and Oct being the most stable among the months contributing to the tensor), this within the CLM5 JULES differentiation (mainly); - in Fig.6 for 2006 a shift from 2005 in concert for all months with a contributing effect; - in Fig.9 despite a very similar pattern to Fig.3, more year to year uncertainties and May differs from Sep with an increasing trend. - in Fig.10 higher variations over the years for the months with positive weighting.

**2: "Another point is related to key objectives and research tasks of the paper. In fact, I cannot clearly see the general study idea and an "internal linkage" of the chapters describing modeling aspects relevant to CSI and NPP predictions. I see some mosaic of separate individual tasks that have to be more closely linked to one another in the paper."**

The reviewer in fact provides a good summary of the concept of the paper in their introductory remarks. As expressed in the abstract and Section 1, its motivation is CSI prediction with predictor variables that are output from LSMs, but it is relevant for any other predictive modelling using LSMs as predictors of inputs to the analysis. The main goal of the paper is then to set out a methodology to quantify and describe the commonalities and uncertainties when using LSMs to provide these predictors. The paper concentrates on proposing a data science method to do this and illustrating the

different possible outcomes of using this method. This is helpful for the CSI community modelling working in CLINF to understand what they can realistically expect from LSMs and the associated uncertainties. Translating this understanding into CSI modelling uncertainty is a next step when their CSI models are better developed.Âǎ

Regarding "internal linkage", we identify on p. 2-3 three questions that we wanted to address in this paper, two concerned with quantifying differences between LSM outputs relevant to CSIs, and one concerned with the impact of the choice of driving GCM on variability in LSM output. The paper is structured to answer these three questions as far as we could with the available data. The "mosaic of separate individual tasks" reflects that different analyses were performed in Sections 4, 5 and 6 to answer these different but related questions which each contributes to understanding the overall variability in LSM predictions of NPP and its partitioning between model and GCM variability.

We've added text to the conclusions about how CSI research in CLINF can benefit from the structured approach to model variability expressed in this paper when considering the uncertainty in CSI predictions under climate change. (*see changes*)

Please also note the supplement to this comment:
https://www.biogeosciences-discuss.net/bg-2019-252/bg-2019-252-AC3-supplement.pdf

Please also note the supplement to this comment:
https://www.biogeosciences-discuss.net/bg-2019-252/bg-2019-252-AC3-supplement.pdf

**Supplement:**

[revised manuscript text omitted]